# The Effect of Breed and Age on the Growth Performance, Carcass Traits and Metabolic Profile in Breast Muscle of Chinese Indigenous Chickens

**DOI:** 10.3390/foods11030483

**Published:** 2022-02-07

**Authors:** Shaolin Deng, Tong Xing, Chunbao Li, Xinglian Xu, Guanghong Zhou

**Affiliations:** 1Key Laboratory of Meat Processing and Quality Control, College of Food Science and Technology, Nanjing Agricultural University, Nanjing 210095, China; dslin@njau.edu.cn (S.D.); chunbao.li@njau.edu.cn (C.L.); xlxus@njau.edu.cn (X.X.); 2College of Animal Science and Technology, Nanjing Agricultural University, Nanjing 210095, China; xingtong@njau.edu.cn

**Keywords:** indigenous chicken, growth performance, carcass trait, age, flavor precursor, metabolomics

## Abstract

Indigenous chickens possess desirable characteristics and account for considerable proportions of the total chicken production in China. The current study examined the growth performance, carcass characteristics and muscle metabolites among a crossbred broiler and two indigenous, yellow-feathered chickens (Mahuang and Tuer) with different ages (60 and 75 days). Results indicated that the crossbred broiler had better feed efficiency, higher breast and thigh muscle yield, as well as a lower abdominal fat percentage than Mahuang and Tuer chickens (*p* < 0.05). Gas chromatography–mass spectrometry-based metabolomics and multivariate analysis revealed sugars, amino acids and organic acids were the predominant metabolites that differed among the three chicken breeds. Growth performance and carcass traits of yellow-feathered chickens exhibited significant differences with the extension of the feeding period (*p* < 0.05). Moreover, differential metabolites reflected altered aminoacyl-tRNA biosynthesis, ATP-binding cassette transporters, pantothenate and CoA biosynthesis, as well as glutathione metabolism in yellow-feathered chickens affected by age. Collectively, this study contributes to a deeper understanding of the production efficiency and chemical composition of precursor flavor in Chinese indigenous, yellow-feathered chicken.

## 1. Introduction

Chicken meat is considered a nutritionally valuable source of high-quality protein, with a lower content of fat and cholesterol than other types of meat. Moreover, chicken meat is not a subject of culturally or religiously set limitations, and it is usually available at affordable prices [1]. These predominant aspects make meat-type chickens the primary resource for humans in most areas of the world [2]. The growing demand for chicken meat has driven the genetic selection of birds towards rapid growth and carcass yield, whereas the spontaneous, idiopathic muscle abnormalities as well as the susceptibility of birds to stress-induced myopathy has also increased [3,4]. In addition, this selection might negatively affect the sensory and functional properties of chicken meat [5].

Broiler production constitutes a significant portion of poultry production, with the global annual production forecasted to be 105.26 million metric tons in 2023 with a predicted growth rate of 1.73% in the years 2019–2023 [6], and it accounts for almost the whole chicken market, particularly in Western countries. However, considerable proportions of chicken meat come from indigenous breeds in the South and Southeast Asia, as well as in some African countries including Ethiopia, Uganda and Kenya [7]. Generally, the meat-type broilers have better production efficiencies than the local breeds, whereas indigenous chickens possess desirable characteristics such as resistance to some diseases and hostile environments, as well as special meat quality traits [8]. For example, Chinese indigenous chickens of four genotypes (Ninghai chicken, frizzle chicken, Ninghai xiang chicken, and Zhenning loquat chicken) showed higher inosine-5′-monophosphate (IMP) content, smaller fiber diameter and a lower shear force value than those of Arbor Acres plus broilers [9]. Similarly, the Korean native chicken received a higher sensory score for chewiness, odor, taste and overall acceptance than that from broilers [10,11]. However, Dyubele et al. [12] observed that consumers gave higher sensory scores, including the initial impression of juiciness, first bite, as well as sustained impression of juiciness for broiler chicken meat than for indigenous chicken meat from South African. Therefore, detailed studies concerning the carcass characteristics, meat quality traits and chemical composition of meat from broilers and indigenous chickens are needed to give a complete presentation for producers and consumers.

Chinese consumers prefer slow-growing quality chickens owing to their umami and aroma, as well as juicy texture. Yellow-feathered chickens are mainly processed to boiled chicken meat or chicken soup, which are popular in the chicken market of China [13]. According to a report of Chinese Chicken Industry Development in 2015, the consumption of meat from yellow-feathered chickens in China has increased to 4445 kilotons, which accounted for 31.8% of the total chicken production [14]. Indigenous, yellow-feathered chicken breeds are mainly concentrated in south China, especially in Guangdong Province. Among which, Mahuang and Tuer chickens are representative of the local chickens from Guangdong Province, and they have different characteristics in appearance, size, plumage and edible quality [15]. Conventionally, chickens with old age are preferred by Chinese consumers when preparing boiled chicken meat or chicken broth as they believe aged chickens can provide more flavor and nutrition. Indeed, significant differences of flavor precursor substances including lactate, creatine, IMP, glucose, carnosine, anserine, taurine and glutamine were observed in Wuding chickens between the ages of 110 and 230 days [16]. However, the extension of the feeding period inevitably impacts the production efficiency of chickens, which represents a commercial disadvantage for producers [17]. Thus, it is necessary to investigate the effects of age on the growth performance and carcass traits of indigenous chicken, instead of only thinking about the deposition of precursors of meat flavor.

Collectively, the objective of the current study was to investigate feed efficiency, carcass characteristics and the compare chemical composition of precursor flavor substances in the breast muscle of the meat-type crossbred broiler and two indigenous, yellow-feathered chickens. Besides, the effects of breed and age on the growth performance, carcass traits and endogenous compounds of indigenous, yellow-feathered chickens were evaluated. The derived information could provide a theoretical basis for popularizing yellow-feathered chicken meat or processed products, as well as for their selection and breeding.

## 2. Materials and Methods

### 2.1. Birds, Husbandry and Experimental Design

All experimental procedures and bird management were approval by the Institutional Animal Care and Use Committee of Nanjing Agricultural University under protocol number SYXK 2017-007. A total of 240 one-day-old male chicks of a commercial meat-type broiler strain and two indigenous, yellow-feathered breeds with similar body weight (provided by the Southern Poultry Breeding Company of Wens Co., Ltd., Yunfu, China) were completely randomly allocated into 5 experimental treatments, with 6 replicate pens per treatment and 8 chicks per pen. These 5 experimental groups included the following: (1) CON group (817 Crossbred broiler raised for 55-d), (2) MH60 group (Wens Yellow-Feathered Mahuang chicken raised for 60-d), (3) MH75 group (Wens Yellow-Feathered Mahuang chicken raised for 75-d), (4) TE60 (Wens Yellow-Feathered Tuer chicken raised for 60-d) and (5) TE75 (Wens Yellow-Feathered Tuer chicken raised for 75-d). Diets were provided in three phases. The experimental diets were formulated to meet the minimum requirements of the National Research Council guidelines (1994). The ingredients and nutrient composition of the basal diet are presented in Table 1. All chickens were allowed ad libitum access to feed and water. Room temperature was kept between 33–35 °C from days 1 to 3 and reduced to a final temperature of 25 °C at the rate of 1 °C every day. A 12 h light–dark cycle was provided throughout the trial.

### 2.2. Growth Performance

After feed deprivation for 12 h, body weight (BW) and feed consumption on a pen basis were recorded at 55 days of age for the CON group, at 60 days of age for the MH60 and TE60 groups, as well as at 75 days of age for the MH75 and TE75 groups, respectively, to calculate the average daily gain (ADG), average daily feed intake (ADFI) and feed:gain ratio (F/G).

### 2.3. Slaughter and Sample Collection

Upon slaughter age, birds were weighted individually, labeled and then euthanized by electrical stunning followed by exsanguination. After bleeding by cutting the carotid arteries and jugular veins, birds were scalded in water at 60 °C for 45 s and introduced into a commercial plucking machine. After chilling in an ice-water bath for 5 min, the defeathered carcasses were weighed to calculate the dressing percentage by dividing the carcass weight by the live BW. The carcasses were eviscerated and the head and feet were discarded, then they were weighed to determine the percentage of the eviscerated yield. The abdominal fat tissue (from the surrounding proventriculus and the gizzard down to the cloaca), breast and thigh muscles were removed and weighed to calculate the proportions of abdominal fat, breast and thigh muscles by dividing the respective weight by the eviscerated weight [18]. Meanwhile, the whole breast muscle (pectoralis major and minor) was wrapped in aluminum foil and stored in liquid nitrogen for further analysis.

### 2.4. Preparation of Muscle Samples for GC-MS Analysis 

The preparation of the muscle sample for gas chromatography–mass spectrometry (GC-MS) analysis was conducted according to Tan et al. [19], with minor modifications. In brief, each freeze-dried powdered breast muscle tissue (30 mg) was added to 20 μL of L-2-chlorophenylalanine (0.3 mg/mL) dissolved in methanol (internal standard) and 600 μL of the extraction solvent with methanol/water (4/1, *v*/*v*). Muscle samples were stored at −20 °C for 2 min and then homogenized at 13,500 rpm for 1 min. Then, 120 μL of chloroform was added to each sample, vortexed and placed at −20 °C for 20 min. After centrifugation at 13,000 rpm for 10 min at 4 °C, 150 μL of the supernatant was vacuum-dried. The quality control (QC) sample was combined by mixing the aliquot of all the individual samples. Subsequently, 80 μL of pyridine containing 15 mg/mL of methoxylamine hydrochloride was added, and the resultant mixture was vortexed for 2 min and incubated at 37 °C for 90 min. Next, 50 μL of N,O-bis-trimethylsilyltrifluoroacetamide with 1% trimethylchlorosilane and 20 μL of n-hexane were further added and then derivatized at 70 °C for 60 min. The samples were placed at room temperature for 30 min before GC-MS analysis.

### 2.5. GC-MS Analysis

The derivatized samples were analyzed on an Agilent 7890B GC system coupled to an Agilent 5977A MSD system (Agilent Technologies Inc., Santa Clara, CA, USA). A DB-5MS fused-silica capillary column (i.d. 30 m × 0.25 mm, firm thickness 0.25 μm, Agilent J & W Scientific, Folsom, CA, USA) was utilized for separation. Helium (>99.999%) was used as the carrier gas at a constant flow rate of 1 mL/min through the column, and the injection port temperature was maintained at 260 °C. The injection volume was 1 μL in splitless mode with a 5 min delay of the solvent. The GC temperature program was set as follows: Initially held at 60 °C for 0.5 min, ramped up to 125 °C at a rate of 8 °C/min, ramped up to 210 °C at a rate of 5 °C/min, then ramped up to 270 °C at a rate of 10 °C/min, to 305 °C at a rate of 20 °C/min and finally held at 305 °C for 5 min. The temperature of MS quadrupole and the ion source was set to 150 °C and 230 °C, respectively. The collision energy was performed with 70 eV. Mass scans were conducted in a full-scan mode (*m*/*z* 50 to 500). The QC samples were injected at regular intervals (every 6 samples) throughout the analytical run, in order to provide a set of data from which repeatability could be assessed.

### 2.6. GC-MS Data Preprocessing and Analyses

After GC-MS analysis, the obtained raw chromatograms were converted to a general format using Analysis Base File Converter software. Then, data were imported into MS-DIAL software for preprocessing. Metabolite annotation was performed through the LUG database (Untarget database of GC-MS from Lumingbio, Shanghai, China). Principle Component Analysis (PCA) was initially carried out to observe the overall distribution among the samples and the stability of the whole analysis process. Then, partial least squares–discriminant analysis (PLS-DA) and orthogonal partial least squares–discriminant analysis (OPLS-DA) were subsequently performed using SIMCA software (version 14.1, Sartorius Stedim Biotech, Umeå, Sweden) to distinguish the metabolites that differ between groups. Additionally, we performed a 7-fold cross-validation and 200-response permutation testing to evaluate the reliability and to avoid overfitting of the OPLS-DA model. Variable Importance of Projection (VIP) values of the metabolites were obtained from the OPLS-DA model. A two-tailed Student’s *t*-test was further used to verify whether the metabolites of difference between groups were significant. Differentially abundant metabolites among the groups were selected with a VIP value > 1.0 and a *p* value < 0.05.

### 2.7. Metabolic Pathway Analysis

The differential metabolites were further identified and validated by searching various public databases, including the Kyoto Encyclopedia of Genes and Genomes (KEGG), PubChem and the Human Metabolome Database (HMDB). MetaboAnalyst 3.0, (http://www.metaboanalyst.ca (accessed on 29 October 2021), which incorporates high-quality KEGG metabolic pathway as the backend knowledge base, was used for the metabolomics pathway analysis as described previously [20]. 

### 2.8. Statistical Analysis

The data of the growth performance and carcass traits were submitted to the analysis of variance (ANOVA) using the general linear model procedure of Statistical Analysis System (SAS 9.0, SAS Institute Inc., Cary, NC, USA). The growth performance data were analyzed with the pen as the experimental unit, and the carcass traits were analyzed with the bird as the experiment unit. The significance of differences among the group means was assessed using Duncan’s multiple range test. All results are presented as means ± standard error (SE), and the significance level was set at *p* < 0.05.

## 3. Results

### 3.1. Growth Performance

As exhibited in Table 2, both Mahuang and Tuer chickens had higher BW than the 817 Crossbred chickens (*p* < 0.05). Compared with the 817 Crossbred chickens, the ADFI and F/G ratio of Mahuang chickens were significantly increased (*p* < 0.05). Of which, the ADFI of the MH60 and MH75 groups was increased by 19.48% and 33.69%, respectively. Tuer chickens had lower ADG and a higher F/G ratio as compared with the 817 Crossbred chickens (*p* < 0.05), of which the ADG of TE60 and TE75 group was decreased by 10.92% and 10.41%, respectively. At the same age, BW, ADFI and ADG of Tuer chickens were lower than those of Mahuang chickens (*p* < 0.05), whereas no significant difference in F/G was observed between these two breeds (*p* > 0.05). In addition, BW, ADFI and F/G of Mahuang and Tuer chickens increased (*p* < 0.05), whereas ADG remained unchanged (*p* > 0.05) with the extension of the feeding period.

### 3.2. Carcass Traits

The carcass traits of the indigenous chickens are indicated in Table 3. We observed that the carcass weights of MH60, MH75 and TE75 chickens were higher than that of the CON chickens (*p* < 0.05). Chickens in the MH75, TE60 and TE75 groups had higher dressing percentages than those in the CON group (*p* < 0.05). The eviscerated yield of Mahuang and Tuer chickens at 60 days of age was lower than 817 Crossbred chickens (*p* < 0.05), and the MH75 group had the highest eviscerated yield among the five groups. The breast muscle and thigh muscle weights of the TE60 chickens were lower than those of the CON chickens (*p* < 0.05). MH75 chickens had higher breast muscle and thigh muscle weights than CON (*p* < 0.05). The abdominal fat weight of the CON chicken was the lowest among the five groups (*p* < 0.05). As for the relative weight, the two indigenous, yellow-feathered chickens had lower breast and thigh muscle percentages, whereas they had higher abdominal fat percentages than those of the 817 crossbred broiler regardless of the feeding period (*p* < 0.05). At 60 days of age, Tuer chickens had a significantly lower carcass weight and thigh muscle weight, whereas they had a higher dressing percentage, eviscerated yield and breast muscle yield than those of Mahuang chickens (*p* < 0.05). At 75 days of age, Tuer chickens had lower carcass weight, eviscerated yield, breast and thigh muscle weight and percentage, as well as a higher abdominal fat percentage than those of Mahuang chickens (*p* < 0.05). With the extension of the feeding period, the carcass weight, dressing percentage, eviscerated yield, breast muscle weight and percentage, thigh muscle weight, as well as abdominal weight of chickens in the MH75 group were increased as compared with those in the MH60 group (*p* < 0.05). Only the thigh muscle and abdominal fat percentage remained unchanged between MH75 and MH60. Compared with the TE60 group, the TE75 group had a higher carcass weight, dressing percentage, eviscerated yield, breast and thigh muscle weight, as well as abdominal fat weight and percentage, whereas it had a lower breast muscle percentage (*p* < 0.05). 

### 3.3. GC-MS Chromatogram Inspection and Metabolites Identification

Representative GC-MS total ion chromatograms of samples from the CON, MH60, MH75, TE60 and TE75 groups are exhibited in Appendix A. Overall, the metabolic profile of the pectoralis major muscle from the five groups showed similar outlines. A total of 312 metabolites were unambiguously detected and assigned. No differences in the component or classification of the metabolites were observed among the chicken breeds and ages. Among the 312 metabolites, 67 were organic acids and derivatives, 52 were organic oxygen compounds, 51 were lipids and lipid-like molecules, 32 were organoheterocyclic compounds, 16 were benzenoids, 13 were nucleosides, nucleotides and analogues, 11 were phenylpropanoids and polyketides, 8 were organic nitrogen compounds, 1 was an alkaloid or derivative, 1 was a homogeneous non-metal compound and 60 compounds were unclassified. The top two intense signals in the five groups were identified as L-lactic acid and ciliatine. In addition, D-mannose, inosine and methylamine were identified as the other three intense signals shared in MH60, MH75 and TE75 groups. Detailed information of the annotated metabolites is listed in Appendix A.

### 3.4. Multivariate Analysis of GC-MS Chromatogram Data

Unsupervised PCA was initially performed to provide the overall distribution and differences of metabolomic profiles among the five groups (Appendix A), and two principle components (PCs) were calculated for the five groups, with 16.9 and 12.9% of the total variation being explained by PC1 and 2, respectively. There was an obvious separation between the CON group and the other four groups. PLS-DA score plots (Appendix A) showed obvious separation between the 817 Crossbred chickens and the indigenous, yellow-feathered chickens. Furthermore, clear separation was observed between the Mahuang and Tuer chickens with different feeding periods (Appendix A). All the samples in the score plots were within 95% of the Hotelling *T*^2^ ellipse.

We further applied the OPLS-DA of the chromatogram data to eliminate noise unrelated to the classifications and to highlight the metabolic changes among groups. As indicated in Figure 1A–D, there was a clear trend of separation between the metabolites of the 817 crossbred chicken and the indigenous, yellow-feathered chickens with different ages. The parameters of the CON vs. MH60 model, R2X (cum) = 0.482, R2Y (cum) = 0.993 and Q2 (cum) = 0.740, the parameters of the CON vs. MH75 model, R2X (cum) = 0.454, R2Y (cum) = 0.998 and Q2 (cum) = 0.875, the parameters of the CON vs. TE60 model, R2X (cum) = 0.445, R2Y (cum) = 0.995 and Q2 (cum) = 0.763 and the parameters of the CON vs. TE75 model, R2X (cum) = 0.531, R2Y (cum) = 0.998 and Q2 (cum) = 0.896, all indicated an acceptable goodness of fit and high values of prediction. As described by Mabuchi et al. [21], R2Y ≥ 0.65 and Q2Y ≥ 0.5 generally indicate a satisfactory ability for quantitative prediction. Furthermore, the validation of permutation testing in which the relationship between actual values (vertical axis) and predicted values (horizontal axis) showed good R2 values indicated the models were reliable and not over-fitted (Appendix A). Furthermore, an obvious separation between the metabolites of Mahuang and Tuer chickens at 60 and 75 days of age, respectively, was observed in Figure 1E–H. Similarly, the parameters of the MH60 vs. TE60 model, R2X (cum) = 0.402, R2Y (cum) = 0.994 and Q2 (cum) = 0.649, the parameters of the MH75 vs. TE75 model, R2X (cum) = 0.442, R2Y (cum) = 0.994 and Q2 (cum) = 0.694, the parameters of the MH75 vs. MH60 model, R2X (cum) = 0.454, R2Y (cum) = 0.998 and Q2 (cum) = 0.583, and the parameters of the TE75 vs. TE60 model, R2X (cum) = 0.431, R2Y (cum) = 0.994 and Q2 (cum) = 0.691, indicated >99% of the samples (data records) fit the established mathematic models and accuracy of prediction. In addition, the validation of permutation testing indicated the models were reliable and not over-fitted (Appendix A). Overall, these findings suggest that the OPLS-DA model could explain and predict differences among the groups.

### 3.5. Differential Metabolites Analysis

According to the selection criteria (VIP > 1 and *p* < 0.05), we identified 54, 69, 55 and 87 differentially abundant metabolites in the MH60, MH75, TE60 and TE75 groups, respectively, as compared with the CON group. Detailed information of these differential metabolites and fold change values are listed in Appendix A. 

Compared with the CON group, the concentrations of 41 metabolites were increased and only 13 metabolites were decreased, respectively, in the MH60 group, of which the increased metabolites of altrose, *N*-acetyl-d-hexosamine, L-sorbose, D-xylulose and D-fructose as well as the decreased metabolites of D-fructose-1-phosphate, 4-aminophenol, 1-kestose, diclofenac and pyruvic acid contributed greatly to the grouping of MH60 and CON. Among the 44 increased and 25 decreased metabolites identified between the MH75 and CON groups, the increased L-sorbose, D-fructose, glucose-1-phosphate, D-tagatose and altrose as well as the decreased beta-alanine, tyramine, 4-aminomethylcyclohexane carboxylic acid, D-fructose-1-phosphate and 2-aminoethyl methacrylate were the major contributors for their grouping. We observed that regardless of the feeding period, Mahuang chickens had higher concentrations of altrose, L-sorbose, D-fructose and lower concentrations of D-fructose-1-phosphate in the breast muscle as compared with the 817 Crossbred chickens.

The concentrations of 27 metabolites were increased and 28 metabolites were decreased, respectively, in the TE60 group as compared with the CON group, of which the increased metabolites of N-acetylgalactosamine, altrose, *N*-acetyl-d-hexosamine, cadaverine and 5,7-dihydroxy-4′-methoxyisoflavone as well as the decreased metabolites of ferulic acid, tyramine, beta-alanine, altro-2-heptulose-7-phosphate and guanine contributed greatly to the grouping of TE60 and CON. Among the 41 increased and 46 decreased metabolites identified between the TE75 and CON groups, the increased L-sorbose, D-tagatose, D-fructose, altrose and *N*-acetyl-d-hexosamine as well as the decreased dehydroascorbic acid, beta-alanine, 4-aminomethylcyclohexane carboxylic acid, tyramine and ferulic acid were the major contributors for their grouping. It was observed that Tuer chickens had higher concentrations of altrose, *N*-acetyl-d-hexosamine as well as decreased concentrations of ferulic acid, tyramine and beta-alanine in the breast muscle as compared with the 817 Crossbred broiler.

Moreover, we conducted a pairwise comparison of the metabolites between the two indigenous, yellow-feathered chickens at two different ages. The results indicated that a total of 47, 52, 73 and 59 different metabolites were observed between the MH60 and TE60 groups, between the MH75 and TE75 groups, between the MH75 and MH60 groups, as well as between the TE75 and TE60 groups, respectively (Appendix A). At 60 days of age, Mahuang chickens had 37 increased and 10 decreased metabolites compared with Tuer chickens, and the increased ferulic acid, norvaline, glycyl proline, alprenolol and D-ribose as well as the decreased N-acetylgalactosamine, butanedioic acid, behenic acid, tetracosanoic acid and gallocatechin were the major contributors for their grouping. At 75 days of age, Mahuang chickens had 45 increased and 7 decreased metabolites compared with Tuer chickens, and the increased metabolites of L-glutamine dehydrated, guanine, trimethoprim, butanedioic acid and 3,5-dimethyl-benzoic acid, as well as the decreased metabolites of quinic acid, spermine, altrose, *N*-acetyl-d-hexosamine and melezitose contributed greatly to this grouping. 

The long feeding period (75 days) resulted in 34 increased and 39 decreased metabolites in the breast muscle of Mahuang chickens as compared with the short feeding period (60 d). The increased D-fructose-1-phosphate, 4-aminophenol, butanedioic acid, 1,2,4-benzenetriol and pyruvic acid as well as the decreased D-ribose-5-phosphate, N-methyldiethanolamine, phthalic acid, altrose and L-arabitol were the major contributors for their grouping. As for the Tuer chickens, 27 increased and 32 decreased metabolites were observed in TE75 compared with TE60. The increased metabolites of methylphosphonic acid, D-fructose, glucose 6-phosphate, L-sorbose and norvaline as well as the decreased metabolites of dehydroascorbic acid, cadaverine, 4-aminomethylcyclohexane carboxylic acid, N-acetylgalactosamine and beta-alanine contributed greatly to their grouping.

### 3.6. Metabolite Set Enrichment Analysis 

The top 20 metabolic pathways related to the discriminating metabolites were identified by the individual OPLS-DA models. The most-impacted pathways (*p* < 0.01) that were identified by the differential metabolites between MH60 and CON were arginine biosynthesis, aminoacyl-tRNA biosynthesis, pentose and glucuronate interconversions, the pentose phosphate pathway, the neuroactive ligand–receptor interaction, ATP-binding cassette (ABC) transporters, alanine, aspartate and glutamate metabolism, pantothenate and CoA biosynthesis, beta-Alanine metabolism, phenylalanine, tyrosine and tryptophan biosynthesis, the gap junction, tyrosine metabolism, as well as butanoate metabolism (Figure 2A). However, only three metabolic pathways including arginine and proline metabolism, beta-Alanine metabolism, as well as D-Arginine and D-ornithine metabolism (Figure 2B) were found to be the most affected (*p* < 0.01) between MH75 and CON. The most responsible pathways (*p* < 0.01) identified by the discriminating metabolites between the TE60 and CON groups were beta-Alanine metabolism, purine metabolism and the citrate cycle (Figure 2C). As for the comparison of TE75 and CON, the most affected pathways were identified as aminoacyl-tRNA biosynthesis, arginine and proline metabolism, beta-Alanine metabolism, glutathione metabolism, purine metabolism, arginine biosynthesis, alanine, aspartate and glutamate metabolism, as well as pantothenate and CoA biosynthesis (Figure 2D). Overall, most of the enriched pathways between the 817 Crossbred chickens and the indigenous, yellow-feathered chickens were related to amino acid or derivative biosynthesis and metabolism, as well as purine metabolism.

For the comparison between Mahuang and Tuer breeds, the most responsible pathways (*p* < 0.01) identified by the discriminating metabolites between the MH60 and TE60 groups were aminoacyl-tRNA biosynthesis, pentose and glucuronate interconversions, the pentose phosphate pathway, phenylalanine metabolism as well as biosynthesis of unsaturated fatty acids (Figure 3A). Aminoacyl-tRNA biosynthesis, alanine, aspartate and glutamate metabolism, arginine and proline metabolism, glycine, serine and threonine metabolism, pantothenate and CoA biosynthesis, beta-Alanine metabolism, glutathione metabolism, as well as tyrosine metabolism were identified between MH75 and TE75 (Figure 3B). Aminoacyl-tRNA biosynthesis was the only pathway altered between Mahuang and Tuer breeds without regard to the feeding period. 

For the comparison of different feeding periods, the most impacted pathways (*p* < 0.01) that were identified between MH75 and MH60 were aminoacyl-tRNA biosynthesis, ABC transporters, arginine and proline metabolism, valine, leucine and isoleucine biosynthesis, the gap junction, glutathione metabolism, ferroptosis, alanine, aspartate and glutamate metabolism, pentose and glucuronate interconversions, pantothenate and CoA biosynthesis, as well as phenylalanine metabolism. The most responsible pathways (*p* < 0.01) identified between the TE75 and TE60 groups were aminoacyl-tRNA biosynthesis, pantothenate and CoA biosynthesis, beta-Alanine metabolism, glutathione metabolism, as well as ABC transporters.

## 4. Discussion

The consumption of chicken meat has been increasing during the past few years. Apart from the fast-growing commercial broiler chickens, indigenous chickens share a considerable proportion of worldwide chicken production, particularly in China. The commercial broiler chickens have superior production traits including faster growth rates and better feed efficiency than indigenous chickens. The selection strategy for broiler chickens is to meet the demand of the ever-increasing world population, whereas the indigenous breeds have not undergone intensive genetic selection [3,22]. Herein, we observed that the F/G ratio in the meat-type strain was lower than that of the two yellow-feathered breeds. Similar to our findings, MaCrea et al. [23] indicated that Delaware chickens showed a higher overall feed conversion ratio as compared with broilers. Interestingly, we noticed that ADFI values of Mahuang chickens were higher than that of the crossbred broiler, and no difference in ADG was observed between these two breeds. This was not in accordance with previous studies, as the commercial broiler generally exhibited higher ADFI and ADG values than that of local breeds [24,25]. In addition, ADG values of the yellow-feathered breeds were much higher than that reported for the other local breeds [22,25]. It is shown that selection or crossbreeding greatly improves the production efficiency of these two indigenous, yellow-feathered chickens while maintaining their original appearance characteristics. There is a close relationship between the duration of rearing and the ultimate economic effect of chicken production [26]. Our results indicated that extending the feeding period from 60 days to 75 days resulted in an increase in the F/G ratio of the two yellow-feathered breeds. This was in accordance with the study of Połtowicz [17], in which the F/G of slow-growing chickens increased from 1.90 to 2.18 when extending the raising time from 56 days to 84 days. 

We observed that the dressing percentage of the meat-type crossbred broiler and two breeds of yellow-feathered chicken in the current study was lower than that reported for Arbor Acres broilers [27]. However, the dressing percentage was higher in Mahuang chickens of 75 days as well as in Tuer chickens of 60 days and 75 days than in meat-type crossbred broilers. Similarly, Dyubele et al. [12] found that the dressing percentage of indigenous chickens from South Africa was higher than Ross^®^ broilers. Crossbred broilers had a higher breast and thigh muscle yield, whereas they had a lower abdominal fat percentage as compared with Mahuang and Tuer chickens, indicating the obvious differences of muscle and fat development between yellow-feathered breeds and the meat-type strain. Differences in the dressing percentage, eviscerated yield and breast muscle yield were observed between Mahuang and Tuer chickens at the same age. Singh et al. [28] compared carcass traits among three Indian indigenous breeds (Aseel, Vanraja and Kadaknath) at 6 weeks of age, and their results indicated that Aseel chickens had the highest dressing percentage, hot and cold carcass weight as well as meat yield, and these traits were comparable between Vanraja and Kadaknath chickens. However, indigenous chicken strains from Thailand (Black-boned and Thai native) at 12 weeks of age exhibited no clear differences in retail cuts with bones and cuts obtained via the Thai cutting style [29]. The dressing percentage and eviscerated yield of Mahuang and Tuer chickens at 75 days of age were higher than that of chickens at 60 days of age. In accordance with this, Shanzhongxian × W-line crossbred chicken had a higher BW and dressing percentage at 180 days of age than those at 70 days of age [30]. We assumed that the higher dressing percentage and eviscerated yield of Mahuang and Tuer chickens might be ascribed to their increased BW, as a high live weight has been associated with high carcass yield [31].

Metabolomics has been widely used to understand various farm animals’ skeletal muscles and meat for the improvement and assessment of meat quality [32]. GC-MS has been used as an untargeted metabolomics approach to identify and characterize small molecules of metabolites in muscle tissues due to its robustness, high sensitivity and capability to provide a snapshot for a broad range of metabolites within a single analysis [19,21]. As expected, comprehensive multivariate analysis revealed that there were significant differences in the concentrations of metabolites including carbohydrates, amino acids, peptides, fatty acids, organic acids and others among the chickens with different breeds and ages in the current study. 

Among these different metabolites, sugars were the major contributors for the differentiation of the meat-type crossbred broiler and yellow-feathered chickens. For example, the concentrations of altrose, *N*-acetyl-d-hexosamine, L-sorbose, D-xylulose, D-fructose, glucose-1-phosphate, D-tagatose and D-arabinose were higher, whereas the concentrations of D-fructose-1-phosphate and 1-kestose were lower in the breast muscle of Mahuang chickens as compared with the crossbred broiler. Similarly, Tuer chickens had higher concentrations of N-acetylgalactosamine, altrose, *N*-acetyl-d-hexosamine, L-sorbose, D-tagatose, D-fructose, altrose, *N*-acetyl-d-hexosamine and glucose-1-phosphate, as well as lower concentrations of D-fructose-1-phosphate and 1-kestose than the crossbred broiler. As one of the most important sources of nonvolatile flavor precursor substances, reducing sugars can react with α-amino acids via the Maillard reaction and promote the formation of the cooked meat aroma compound [33]. Furthermore, the addition of reducing sugars could induce protein glycation and facilitate the solubilization of myosin, which in turn might increase the water holding capacity of meat [34]. Therefore, the increased concentrations of reducing sugars in the yellow-feathered chicken might contribute to its flavor and cooking yield. Based on the KEGG pathway database, we found beta-Alanine metabolism was altered in all yellow-feathered chickens with different ages as compared with the crossbred broiler. Among the identified metabolites involved in this pathway, beta-alanine and quinolinic acid were found to be lower in MH and TE groups than the CON. 

Higher concentrations of amino acids including L-glutamic acid, L-aspartic acid, L-phenylalanine and L-tyrosine were found to be involved in the most impacted metabolic pathways between MH60 and CON. Amino acids have been indicated to be the main nonvolatile components and important precursors of volatile components [35]. In addition, free amino acids can enhance the taste of meat. Thus, the higher concentrations of L-glutamic acid and L-aspartic acid might increase the umami taste of Mahuang chicken [36], which is a very important component for chicken meat. 

Tuer chickens showed decreased concentrations of ferulic acid, tyramine and beta-alanine compared to those in the crossbred broiler. Dietary supplementation of ferulic acid has been demonstrated to improve meat quality and feed efficiency owing to its antioxidant capacity against lipid peroxidation and possible effect on muscle growth [37]. Moreover, the dietary mixture feed additive of ferulic acid and probiotics can promote the accumulation of umami nucleotides, ketones and esters [38]. Tyramine is often produced by the decarboxylation of tyrosine during fermentation or storage in various foods including cheese, soy sauce, sauerkraut and processed meat. A high dietary intake of tyramine can cause a rise in systolic blood pressure by at least 30 mm Hg that may lead to hypertensive crises [39]. Furthermore, purine metabolism was predicted to be different between the Tuer chicken and the crossbred broiler. Among the metabolites involved in this pathway, hypoxanthine and xanthosine exhibited higher concentrations in the breast muscle of Tuer chickenz. As described by Ichimura et al. [40], hypoxanthine can enhance the overall taste of cured meat, though it exhibits a bitter taste. 

Amino acids were found to exert obvious different patterns between Mahuang and Tuer chickens. At 60 days of age, Mahuang chickens had higher concentrations of L-methionine, L-phenylalanine, L-tyrosine, Leucine, L-asparagine, L-phenylalanine and L-tyrosine, as well as a lower concentration of L-histidine than those in Tuer chickens. At 75 days of age, higher concentrations of glycine, L-glutamine, leucine, L-proline, L-asparagine and L-valine were found in Mahuang chickens as compared with Tuer chickens. As aforementioned, the differential amino acid profile between these two breeds may contribute to their distinct flavor [35,36]. The concentrations of several sugars including D-ribulose 5-phosphate, D-xylulose, L-threo-2-pentulose and D-ribose were higher in Mahuang chickens than Tuer at 60 days of age. They were identified to be involved in pentose and glucuronate interconversions as well as the pentose phosphate pathway. Pentoses, particularly ribose from meat ribonucleotides and the sulfur-containing amino acid, are important precursors for the Maillard reaction and caramelization, providing abundant volatile compounds of cooked meat [41]. 

We observed that age had significant impacts on the accumulation of metabolites in the breast muscle of yellow-feathered chickens. Additionally, metabolic pathways of aminoacyl-tRNA biosynthesis, ABC transporters, pantothenate and CoA biosynthesis, as well as glutathione metabolism were involved in the metabolic response to different ages in both Mahuang and Tuer chickens. Amino acids or their derivatives were the dominant metabolites involved in these pathways, and concentrations of almost all of the amino acids were decreased in 75-day-old yellow-feathered chickens than those in 60-day-old chickens. For example, L-glutamic acid, L-lysine, L-phenylalanine, L-tyrosine, leucine, L-proline, L-valine, L-threonine, L-isoleucine, L-histidine, L-asparagine and O-phosphoserine involved in aminoacyl-tRNA biosynthesis were lowered with the extension of the feeding period. Similarly, L-glutamic acid, L-lysine, L-valine, L-cystine, 4-hydroxyproline and L-histidine were decreased with the extension of the feeding period. The decrease in these flavor precursors may influence the taste and flavor of chicken meat. IMP is the most important component of umami in the nucleotides of chicken meat [42], and it is an intermediate metabolite of amino acid biosynthesis and metabolism. The lower concentrations of amino acids or their derivatives may also affect the accumulation of IMP in chicken muscle. Moreover, considerable organic acids and derivatives such as butanedioic acid, pyruvic acid, L-2-hydroxyglutaric acid, D-myo-inositol 4-phosphate, N-acetylaspartic acid and aminoadipic acid were observed to exhibit different patterns in the breast muscle of yellow-feathered chickens with different ages. As organic acids are important non-volatile constituents of fresh meat, their significant alterations might also contribute to the change of basic tastes of cooked meat [33]. However, the relationship between the precursor flavor substances and the cooked chicken meat flavor is quite complicated, which needs to be further clarified.

## 5. Conclusions

The results from the current study indicated that the growth performance, carcass traits and metabolic composition of chicken muscle differed among breeds. The crossbred broiler had better feed efficiency, higher breast and thigh muscle yield, as well as a lower abdominal fat percentage compared to yellow-feathered chickens. GC-MS-based metabolomics coupled with multivariate statistical data analysis revealed significant differences in the concentrations of flavor precursors including sugars, amino acids and organic acids among crossbred broiler, Mahuang and Tuer chickens. In addition, we observed differences in the growth performance, dressing percentage and eviscerated yield of yellow-feathered chickens between the age of 60 and 75 days. Aminoacyl-tRNA biosynthesis, ABC transporters, pantothenate and CoA biosynthesis, as well as glutathione metabolism were the main metabolic pathways affected by age in Mahuang and Tuer chickens. In general, this study provides valuable information for recognizing the production efficiency and chemical composition of precursor flavor substances in thee breast muscle of indigenous, yellow-feathered chickens in China. Further studies are needed to assess the meat quality, processing characteristics and aroma profiles of yellow-feathered chickens for their development.

## Figures and Tables

**Figure 1 foods-11-00483-f001:**
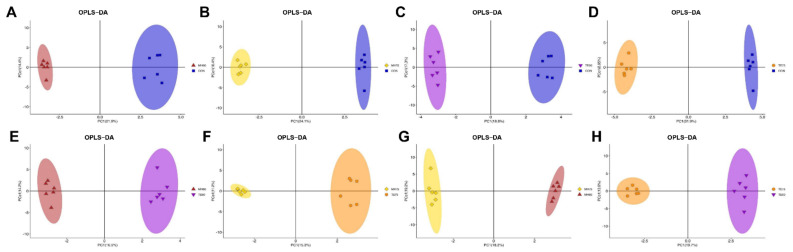
Score plots of orthogonal projections to latent structures discriminant analyses (OPLS−DA) derived from the GC−MS profiles of muscle samples obtained from the MH60 group vs. the CON group (**A**), the MH75 group vs. the CON group (**B**), the TE60 group vs. the CON group (**C**), the TE75 group vs. the CON group (**D**), the MH60 group vs. the TE60 group (**E**), the MH75 group vs. the TE75 group (**F**), the MH75 group vs. the MH60 group (**G**) and the TE75 group vs. the TE60 group (**H**).

**Figure 2 foods-11-00483-f002:**
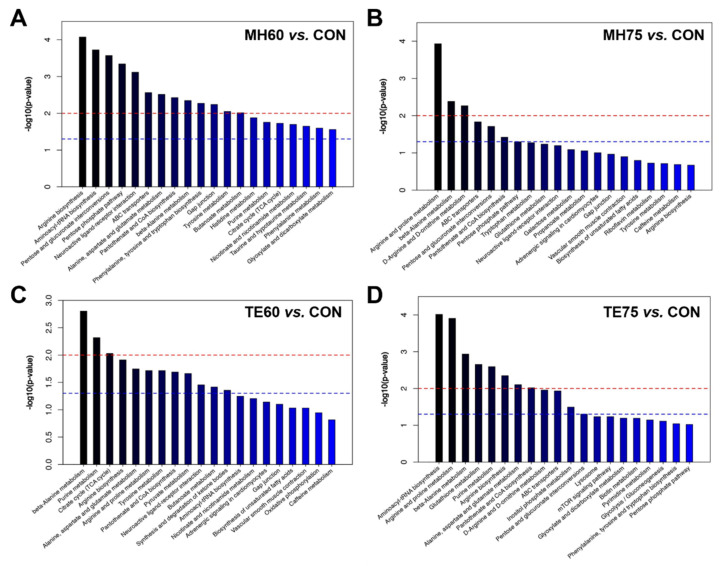
Metabolite set enrichment analysis was performed using all of the discriminating metabolites identified by the four individual orthogonal projections to latent structures discriminant analyses (OPLS-DA) models. The top 20 metabolic pathways were obtained from the MH60 group vs. the CON group (**A**), the MH75 group vs. the CON group (**B**), the TE60 group vs. the CON group (**C**) and the TE75 group vs. the CON group (**D**). The ordinate represents -log(*p*). Bars that exceed the blue and red dotted lines indicate pathways with *p* < 0.05 and *p* < 0.01, respectively.

**Figure 3 foods-11-00483-f003:**
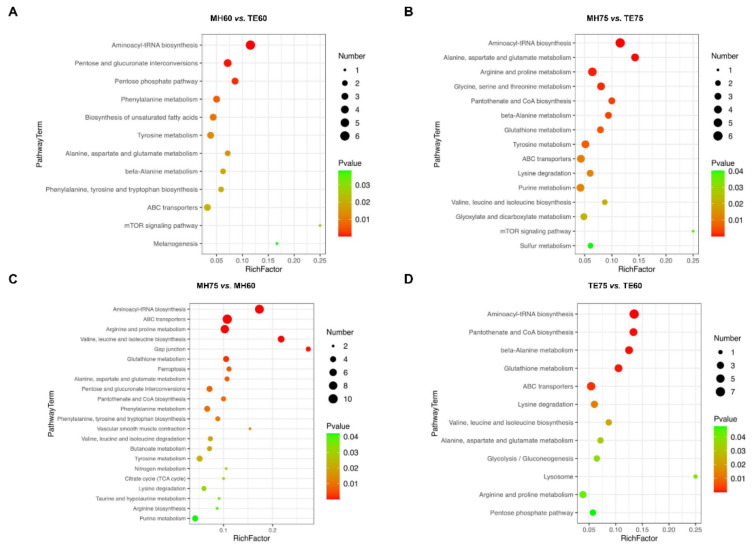
Metabolite set enrichment analysis was performed using all of the discriminating metabolites identified by the individual orthogonal projections to latent structures discriminant analyses (OPLS-DA) models. The most impacted pathways (*p* < 0.05) were obtained from the MH60 group vs. the TE60 group (**A**), the MH75 group vs. the TE75 group (**B**), the MH75 group vs. the MH60 group (**C**) and the TE75 group vs. the TE60 group (**D**). The size of the dot represents the metabolite number, and the color indicates the degree of statistical significance.

**Table 1 foods-11-00483-t001:** Ingredients and nutrient composition of the basal diet.

Ingredient	Starter ^1^	Grower ^2^	Finisher ^3^
Corn	618.99	677.71	711.48
Soybean meal (46%)	229.89	140.69	100
Cottonseed meal (46%)	50	60	60
Corn gluten meal (60%)	50	57.12	61.2
Soybean oil	10.2	21	26.6
Limestone	13.7	14.3	14.3
Dicalcium phosphate	12.4	10.3	8.7
Sodium chloride	3.5	3.5	3.6
Choline Chloride (60%)	0.8	0.5	0.3
Premix ^4^	4	4	4
Lysine (70%)	4.49	6.86	6.83
Methionine (98%)	1.29	2.38	1.46
Threonine (98%)	0.34	1.24	1.13
Mold inhibitor	0.4	0.4	0.4
Calculated nutrient content			
Metabolisable energy (kcal/kg)	2900	3030	3100
Crude protein	20.5	18	17
Calcium	0.9	0.85	0.8
Phosphorus	0.59	0.53	0.49
Available phosphorus	0.351	0.31	0.28
Lysine	1.05	0.97	0.88
Methionine + cysteine	0.71	0.76	0.65
Threonine	0.65	0.63	0.58
Arginine	1.22	1.02	0.92

^1^ For 817 crossbred broiler, the starter phase is 1–21 d. For Wens yellow-feathered Mahuang and Tuer chickens, the starter phase is 1–25 d. ^2^ For 817 crossbred broiler, the grower phase is 22–42 d. For Wens yellow-feathered Mahuang and Tuer chickens, the grower phase is 26–55 d. ^3^ For 817 crossbred broiler, the finisher phase is 43–55 d. For Wens yellow-feathered Mahuang and Tuer chickens, the finisher phase is 56–75 d. ^4^ Premix provided per kilogram of diet: Cu, 5.00 mg; Fe, 69.00 mg; Zn, 84.00 mg; Mn, 98.6 mg; I, 1.14 mg; Se, 0.30 mg; vitamin A (retinyl acetate), 15,000 IU; vitamin D (cholecalciferol), 3000 IU; vitamin E (dl-α-tocopheryl acetate), 25.5 IU; vitamin K3, 2.1 mg; vitamin B1, 2.4 mg; vitamin B2, 9 mg; vitamin B6, 5.1 mg; vitamin B12, 0.02 mg; Calpan, 12 mg; niacin, 48 mg; folic acid, 1.2 mg; biotin, 0.06 mg; Roxarsone, 50 mg; salinomycin, 90 mg.

**Table 2 foods-11-00483-t002:** Effect of breed and age on the growth performance of Chinese indigenous chickens ^1^.

Items	Groups ^2^					SEM	*p* Value
	CON	MH60	MH75	TE60	TE75		
BW (g)	1.99 d	2.17 c	2.72 a	1.91 d	2.40 b	0.03	<0.001
ADFI (g/bird/day)	70.26 c	83.95 b	93.93 a	73.40 c	81.99 b	1.20	<0.001
ADG (g/bird/day)	35.53 a	36.02 a	36.27 a	31.65 b	31.83 b	0.39	<0.001
F/G (g/g)	1.98 c	2.33 b	2.59 a	2.32 b	2.58 a	0.02	<0.001

^1^ The results are represented as the mean and SEM (*n* = 6). Different letters (a–d) in a line indicate significant differences (*p* < 0.05). ^2^ CON, 817 Crossbred chicken raised for 55 d; MH60 and MH75, Wens Yellow-Feathered Mahuang chicken raised for 60 and 75 d; TE60 and TE75, Wens Yellow-Feathered Tuer chicken raised for 60 and 75 d.

**Table 3 foods-11-00483-t003:** Effect of breed and age on the carcass traits of Chinese indigenous chickens ^1^.

Items	Groups ^2^					SEM	*p* Value
	CON	MH60	MH75	TE60	TE75		
Carcass weight (g)	1798.00 d	1952.43 c	2511.04 a	1741.21 d	2214.98 b	23.58	<0.001
Dressing percentage (%)	90.31 c	90.05 c	92.19 a	91.05 b	92.23 a	0.13	<0.001
Eviscerated yield (%)	79.86 b	77.87 d	81.14 a	78.93 c	79.75 b	0.18	<0.001
Breast muscle weight (g)	272.76 b	194.46 d	286.61 a	194.14 d	227.39 c	3.44	<0.001
Breast muscle (%)	17.16 a	11.53 d	12.96 b	12.86 b	11.88 c	0.12	<0.001
Thigh muscle weight (g)	313.86 c	313.86 c	417.73 a	276.73 d	343.30 b	5.07	<0.001
Thigh muscle (%)	19.73 a	18.60 cd	18.90 b	18.33 cd	17.93 d	0.17	<0.001
Abdominal fat weight (g)	27.14 c	41.21 b	56.97 a	37.72 b	57.25 a	1.91	<0.001
Abdominal fat (%)	1.67 c	2.37 b	2.51 b	2.43 b	2.89 a	0.09	<0.001

^1^ The results are represented as the mean and SEM (*n* = 48). Different letters (a–d) in a line indicate significant differences (*p* < 0.05). ^2^ CON, 817 Crossbred chicken raised for 55 d; MH60 and MH75, Wens Yellow-Feathered Mahuang chicken raised for 60 and 75 d; TE60 and TE75, Wens Yellow-Feathered Tuer chicken raised for 60 and 75 d.

## Data Availability

Data is contained within the article and Appendix A.

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
