# Peer review of "The Effect of Breed and Age on the Growth Performance, Carcass Traits and Metabolic Profile in Breast Muscle of Chinese Indigenous Chickens"

_foods, 2022, doi:10.3390/foods11030483_

Round 1

Reviewer 1 Report

General comments:

The manuscript is well-written for the most part. These are only a few minor comments to address. The manuscript contains several important findings to draw conclusions. However, this article seems more suitable for the category of journals in animal sciences. Moreover, considering the aim of this study, it is necessary to measure meat and organoleptic quality traits. If these characteristics had been analyzed, it would have been possible to understand more precisely the relationship between precursor flavor substances and eating quality. Why did not the authors conduct the measurements of these characteristics? And, if possible, please provide the carcass characteristics, including carcass, muscle, and fat weights.

Specific comments:

Line 43: change “broiler chickens” to “broilers”.

Line 48: smaller fiber diameter

Line 48: shear force value

Line 57: provide detailed information on special requirements.

Line 213: why did not the authors provide the live, carcass, and muscle (breast and thigh) weights?

Author Response

Reviewer #1:

General comment

  1. The manuscript is well-written for the most part. These are only a few minor comments to address. The manuscript contains several important findings to draw conclusions. However, this article seems more suitable for the category of journals in animal sciences. Moreover, considering the aim of this study, it is necessary to measure meat and organoleptic quality traits. If these characteristics had been analyzed, it would have been possible to understand more precisely the relationship between precursor flavor substances and eating quality. Why did not the authors conduct the measurements of these characteristics? And, if possible, please provide the carcass characteristics, including carcass, muscle, and fat weights.

Response: Thanks for your comments. The objective of this study was to investigate feed efficiency, carcass characteristics and compare chemical composition of precursor flavor substance in breast muscle of meat-type crossbred broiler and two indigenous yellow-feathered chickens in order to provide theoretical basis for popularizing yellow-feathered chicken meat or processed products, as well as for their selection and breeding. The non-volatile constituents of muscle are essential flavor precursors and contribute to the taste of cooked meat as a function of thermally induced reactions (Khan et al., 2015). Characterizing the chemical composition of precursor flavor substance in muscle could provide clues to understand the role and underlying mechanism of metabolites in the development of meat. As you mentioned, sensory evaluation has been widely used to assess poultry meat quality, and understanding the sensory characteristics is crucial in developing new products, markets and evaluating the quality of products (Sirangelo, 2019). Therefore, there is a need to evaluate sensory characteristics and their relationships with physical meat quality traits and the quantity as well as composition of metabolites in breast muscle of chicken. According to your suggestions, we will conduct these measurements in further studies. Per suggestion, we have provided the live body weight and carcass, muscle, as well as fat weights in Table 2 and 3 of revised manuscript.

Specific comments

  1. Line 43: change “broiler chickens” to “broilers”.

Response: Per suggestion, we have corrected accordingly (Line 48 of revised manuscript).

  1. Line 48: smaller fiber diameter

Response: Per suggestion, we have corrected accordingly (Line 53 of revised manuscript).

  1. Line 48: shear force value

Response: Per suggestion, we have corrected accordingly (Line 53 of revised manuscript).

  1. Line 57: provide detailed information on special requirements.

Response: Thanks for this comment. We have revised this sentence in Line 62-64 of revised manuscript.

  1. Line 213: why did not the authors provide the live, carcass, and muscle (breast and thigh) weights?

Response: Thanks for your comments. We have provided the live body weight and carcass, muscle, as well as fat weights in Table 2 and 3 of revised manuscript. 

Reviewer 2 Report

Dear Editor,

thank you for giving me the possibility to review this research paper. It is interesting but I think must be strongly improved.

Below some suggestion for the Authors

Introduction

The introduction section is well written but needs to be improved for references regarding concepts about market (reference number 6 was of 22 years ago and also reference number 2 should be checked for more recent one).

Materials and methods

L96 – why did you report the three phases, are there differences?

L113-116 Please, you should be clearer. When did you measure these parameters?

L185-193 the statistical analysis is the major concern. It is not clear where did you describe the different age at slaughter considered. If I understand you have a CON group at one age of a genetic type, then one genetic group with 2 different ages at slaughter and another one genetic group with two ages at slaughter. If yes, the model you describe (you did not report it) is not correct. The two ages are not represented in the CON group (that had another age at slaughter). Maybe you can consider the 5 groups as different groups, but you can’t consider the binary interaction of two different fixed effects.

Report the model you used, revise your tables because are not clear in the first two columns.

After this maybe your experimental design will change (and consequently your discussion). Before this The discussion section can’t be considered for reviewing process.

Author Response

Reviewer #2:

  1. Introduction-The introduction section is well written but needs to be improved for references regarding concepts about market (reference number 6 was of 22 years ago and also reference number 2 should be checked for more recent one).

Response: Thanks for your suggestion. We have searched the related literatures published in recent years and revised in Line 35, 40-44, 596-597 and 602-604 of revised manuscript.

  1. Materials and methods-L96 – why did you report the three phases, are there differences?

Response: Thanks for this comment. Chickens have different nutritional requirements during their growth periods (Shariatmadari, 2012). The three-phase diets used for the production of chickens in the current study were proved to be suitable for the maximal growth by the Southern Poultry Breeding Company of Wens Co. Ltd.

  1. L113-116 Please, you should be clearer. When did you measure these parameters?

Response: Thanks for this comment, we have added the time for measurements of growth performance in Line 122-123 of revised manuscript.

  1. L185-193 the statistical analysis is the major concern. It is not clear where did you describe the different age at slaughter considered. If I understand you have a CON group at one age of a genetic type, then one genetic group with 2 different ages at slaughter and another one genetic group with two ages at slaughter. If yes, the model you describe (you did not report it) is not correct. The two ages are not represented in the CON group (that had another age at slaughter). Maybe you can consider the 5 groups as different groups, but you can’t consider the binary interaction of two different fixed effects.

Response: Thanks for this comment. We have revised the statistical analysis method and reanalyzed the data according to your suggestion (Line 192-202 of revised manuscript). Also, the results were re-described in Line 205-218 and 227-258 of revised manuscript.

  1. Report the model you used, revise your tables because are not clear in the first two columns.

Response: Per suggestion, we have reported the model used in this study and revised the tables using the revised model (Line 193-202, Table 2 and 3 of revised manuscript).